# CoReVe: Mitigating Object Hallucinations in Large Vision-Language Models via Chain-of-Region Verification

## Abstract

Large vision-language models (LVLMs) have demonstrated impressive performance in various multimodal understanding and reasoning tasks. However, they still struggle with object hallucinations, *i.e.*, the claim of nonexistent objects in the visual input. To address this challenge, we propose **Chain-of-Re**gion **Ve**rification (**CoReVe**), a region-aware visual chain-of-verification method to mitigate object hallucinations in LVLMs in a post-hoc manner. Motivated by how humans comprehend intricate visual information—often focusing on specific image regions or details within a given sample—we elicit such region-level processing from LVLMs and use it as a chaining cue to detect and mitigate object hallucinations. Specifically, our CoReVe consists of six steps: initial response generation, entity extraction, coordinate generation, region description, verification execution, and final response generation. As a simple yet effective method, CoReVe can be seamlessly integrated into various LVLMs in a training-free manner and without relying on external detection models. Extensive experiments on four hallucination benchmarks across four LVLMs demonstrate that CoReVe can significantly alleviate hallucinations in LVLMs. Code will be released to facilitate future research.

## 1 Introduction

Empowered by large language models (LLMs) (Brown et al., 2020; OpenAI, 2023; Dubey et al., 2024; Team et al., 2024; Meta, 2025; Lu et al., 2024; Team et al., 2025), large vision-language models (LVLMs) (Alayrac et al., 2022; Li et al., 2022; Ye et al., 2023; Liu et al., 2023; Zhu et al., 2023; Li et al., 2023a; Dai et al., 2023; Bai et al., 2023; Lu et al., 2024; Chen et al., 2024d) have made significant strides, exhibiting impressive multimodal understanding and reasoning capabilities (Thrush et al., 2022; Chen et al., 2024a; Kuckreja et al., 2024). Despite their remarkable advancements, existing LVLMs still suffer from *object hallucinations*—producing objects that do not exist in the given image (see Figure 1(a)). This issue has been an Achilles' heel that hinders the broader applications of LVLMs in real-world scenarios.

To mitigate object hallucinations in LVLMs, existing works have primarily focused on three strategies: (i) instruction fine-tuning (Ouyang et al., 2022; Liu et al., 2024a; Gunjal et al., 2024; Wang et al., 2024; Lee et al., 2024), (ii) decoding process optimization (Huang et al., 2024; Leng et al., 2024; Liu et al., 2024b), and (iii) integration of external expert models (Manakul et al., 2023; Kar et al., 2024; Yin et al., 2024; Zhou et al., 2024b; Wu et al., 2024; Shi et al., 2025). These approaches have achieved substantial progress in reducing hallucinations in LVLMs, laying important foundations for improving their reliability. However, they inevitably have the following drawbacks: (i) demands significant computational resources, (ii) requires access to internal model parameters, and (iii) heavily relies on multiple external expert models.

Previous studies in LLMs focusing on text tasks have demonstrated that techniques such as chain-of-thought (Wei et al., 2022; Dhuliawala et al., 2024) can effectively reduce LLM hallucinations, leading to increased reliability of results by guiding them through a structured thought process. In this paper, we are interested in investigating whether such a chaining process is helpful to alleviate LVLM hallucinations. Unlike LLMs, which primarily process and interpret text, the key challenge in LVLMs lies in how to comprehend and reason about the visual information presented in images.

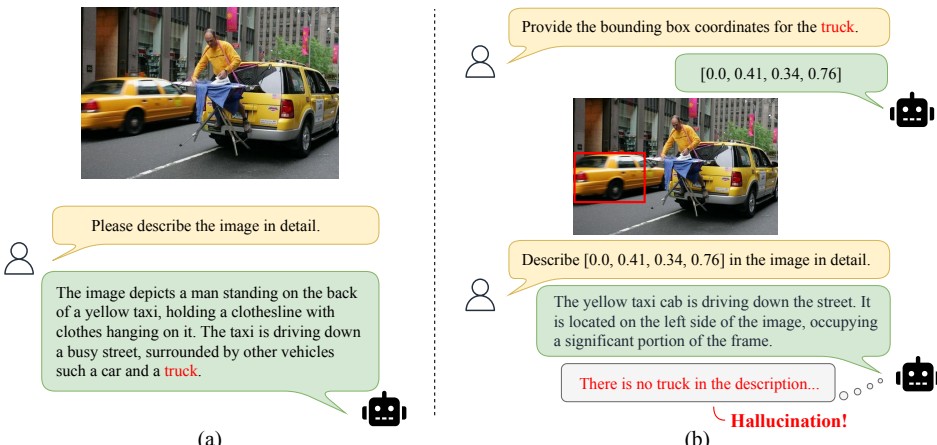

Figure 1: (a) An example of object hallucinations in LVLMs with the hallucinated object highlighted in red. (b) By eliciting region-level processing from LVLMs and using it as a chaining cue, we can detect and mitigate object hallucinations.

To tackle this challenge, we propose **C**hain-**o**f-**Re**gion **Ve**rification (**CoReVe**), a *training-free* method that can directly correct the hallucinations of LVLMs in a post-hoc manner. Our method is inspired by how humans comprehend intricate visual information: we often focus on specific image regions or details within a given sample. As illustrated in Figure 1(b), to elicit such region-level processing from LVLMs, we first obtain the bounding box coordinates for the object of interest by inquiring the LVLM. We then prompt the model to provide a detailed description of the region based on the provided coordinates. Whether the region description contains the original object or not serves as an indicator of a possible hallucination.

Specifically, our CoReVe performs six core steps: (1) *Initial response generation* generates the initial response using the LVLM, given a query (*i.e.*, a question and an image); (2) *Entity extraction* extracts entities in the response; (3) *Coordinate generation* generates region coordinates for each entity; (4) *Region description* describes each region with coordinates in detail; (5) *Verification execution* checks whether the region descriptions contain the original entities to check for inconsistencies or mistakes; and (6) *Final response generation* generates a revised response incorporating the verification results, given the discovered inconsistencies.

We evaluate the effectiveness of CoReVe on several widely used hallucination benchmarks: POPE (Li et al., 2023b) and MME (Fu et al., 2023) for closed VQA tasks, as well as CHAIR (Rohrbach et al., 2018) and GPT-4o assisted evaluation for open-ended generation tasks. Our extensive experiments across multiple advanced LVLMs demonstrate that CoReVe consistently mitigates hallucinations in diverse evaluation settings.

Our main contributions are summarized as follows:

**1)** We introduce CoReVe, a region-aware visual chain-of-verification method to mitigate object hallucinations in LVLMs. Our simple yet effective plug-and-play method can be seamlessly applied to various LVLMs without requiring retraining or external detection models.

**2)** We contribute a comprehensive study on how to design visual prompts to better incorporate the visual information into CoReVe, which has largely been ignored in existing language-centric hallucination correction methods.

**3)** We conduct extensive experiments across multiple LVLMs and evaluation settings. CoReVe achieves consistent and significant performance improvements, demonstrating its strong potential to alleviate hallucinations. We also show that CoReVe achieves the best trade-off between performance and test-time compute compared to prior state-of-the-art post-hoc correction work.

## 2 RELATED WORK

**Large Vision-Language Models.** Large language models (LLMs) (Brown et al., 2020) have transformed the field of natural language processing, de-facto replacing other solutions for many tasks.

Recent commercial models (OpenAI, 2023; Team et al., 2024) have achieved remarkable performance across most text benchmarks, with open-weight solutions closely tracking their performance with a delay of only few months (Dubey et al., 2024; Lu et al., 2024; Meta, 2025; Team et al., 2025). Building on the success of these purely textual models, the community came up with ways of interfacing them with visual encoders to allow the model to operate on multimodal inputs (*e.g.*, image and text). Early examples of this category of models, called large vision-language models (LVLMs) (Alayrac et al., 2022; Li et al., 2022), were highly successful in paving the way to train truly multimodal language models like recent commercial models (Jaech et al., 2024; Team et al., 2024). Building on top of open-weight models, open-source LVLMs have also been developed (Ye et al., 2023; Liu et al., 2023; Zhu et al., 2023; Li et al., 2023a; Dai et al., 2023; Bai et al., 2023; Lu et al., 2024; Chen et al., 2024d). All models share a similar architecture with a dedicated visual encoder to transform images into a latent representation that can later be consumed by an LLM together with an additional text input.

**Hallucination in LVLMs.** In the context of LVLMs, "hallucinations" (Rohrbach et al., 2018) refer to the generation of textual content that is inconsistent with, or entirely disconnected from, the provided visual information. This can manifest in various ways, including the description of nonexistent objects in an image (object hallucination), the attribution of incorrect properties or characteristics to existing objects (attribute hallucination), or the misinterpretation of relationships and interactions between visual elements. This has been a field of intense research with the publication of several benchmarks to be able to measure the ability of models not to hallucinate (Li et al., 2023b; Fu et al., 2023; Wang et al., 2023; Xu et al., 2024; Lovenia et al., 2024; Chen et al., 2024b; Sun et al., 2024).

Several methods have been proposed to mitigate hallucinations in LVLMs. These can be broadly categorized in: (i) methods relying on careful instruction tuning (Ouyang et al., 2022) to try to teach the model not to make facts up (Liu et al., 2024a; Gunjal et al., 2024; Wang et al., 2024; Lee et al., 2024; Zhou et al., 2024a; Yu et al., 2025), (ii) methods proposing customized decoding strategies to mitigate the likelihood of hallucinations (Huang et al., 2024; Leng et al., 2024; Chen et al., 2024c; Liu et al., 2024b), (iii) methods relying on the integration of additional encoders to mitigate hallucinations (Kar et al., 2024; Shi et al., 2025), and (iv) methods relying on expert models to detect, verify and possibly fix hallucinations in a self-criticism loop (Manakul et al., 2023; Yin et al., 2024; Zhou et al., 2024b; Wu et al., 2024), similar to a chain-of-thought loop (Wei et al., 2022). Among the four strategies, the last one has been most successful. We follow a similar path and rely on a visually guided chain-of-region verification to reduce hallucinations. Crucially, our chain is truly multimodal, relying on feeding different versions of the same image at different stages of the verification pipeline in a visual chain-of-thought manner (Rose et al., 2023; Zhang et al., 2024).

## 3 Chain-of-Region Verification

Our chain-of-region verification (CoReVe) is a training-free post-hoc correction method to mitigate object hallucinations in LVLMs. The overall pipeline of CoReVe is illustrated in Figure 2. It contains six stages: initial response generation, entity extraction, coordinate generation, region description, verification execution, and final response generation. We detail each stage as follows.

**Stage 1: Initial Response Generation.** Given a query (*i.e.*, a question and an image), we generate the initial response using the LVLM. This first stage also serves as the baseline we wish to improve in our experiments, *i.e.*, we will directly compare this baseline response with our final verified response. Typically, such initial generations are prone to hallucinations. Our CoReVe attempts to identify and correct these hallucinations, as described in the following stages.

**Stage 2: Entity Extraction.** Based on the initial response generated from the LVLM, we extract candidate entities in this stage for subsequent querying and checking, which are typically the objects mentioned in the response. Specifically, following Wu et al. (2024), we adopt an LLM (*i.e.*, GPT-3.5[1]) to complete this task for simplicity and versatility.[2] We denote the set of extracted entities as $\mathcal{E} = \{e_i\}, i = 1, ..., N$, where $N$ is the number of candidate entities. We provide the detailed prompt we use in Appendix C.1.

---

[1] https://platform.openai.com/docs/models/gpt-3.5-turbo

[2] Note that we use GPT-3.5 here for a fair comparison with previous methods. In Section 4.4, we show that CoReVe can still achieve decent results without relying on any other models.

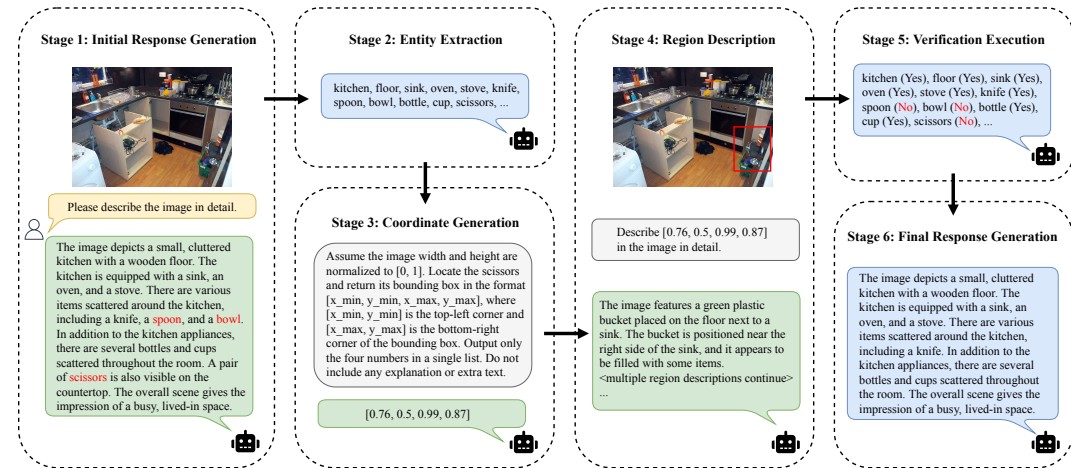

Figure 2: **Overview of our CoReVe pipeline.** CoReVe consists of six stages to mitigate object hallucinations: initial response generation, entity extraction, coordinate generation, region description, verification execution, and final response generation.

**Stage 3: Coordinate Generation.** The primary stage in our CoReVe pipeline involves eliciting region-level knowledge from LVLMs for verification. To achieve this, we prompt the LVLM to provide the position of each extracted entity in the image, using the form of bounding box coordinates. Specifically, given the query image, we prompt the LVLM with the following instruction template: "*Assume the image width and height are normalized to [0, 1]. Locate the {entity} and return its bounding box in the format [x_min, y_min, x_max, y_max], where [x_min, y_min] is the top-left corner and [x_max, y_max] is the bottom-right corner of the bounding box. Output only the four numbers in a single list. Do not include any explanation or extra text.*" This instruction allows us to obtain the bounding box coordinates for each entity $e_i$ from the LVLM. Note that the bounding box coordinates generated in this stage can be inaccurate, *i.e.*, the regions do not necessarily contain the entities due to nonexistence. In the subsequent stages, we will use this as a cue for region-level verification.

**Stage 4: Region Description.** Let $\mathcal{R} = \{r_i\}, i = 1, ..., N$ denote the set of generated region coordinates. In this stage, we aim for region-level verification questions conditioned on $\mathcal{R}$. Specifically, we prompt the LVLM to describe each specific region in the image independently, using the following template: "*Describe {coordinate} in the image in detail.*", where {coordinate} is one of the $r_i \in \mathcal{R}$. We repeat this step multiple times for each region to produce a set of diverse answers, which proves to be beneficial for performance improvements, as shown in the experimental section.

To better instruct the LVLM to focus on local regions, we also explore several ways to provide the image content to the LVLM. Specifically, for each entity $e_i$, we consider the following forms of image prompts: (i) the original image, (ii) the image overlaid with the bounding box coordinates $r_i$, and (iii) a crop from the original image using the bounding box coordinates $r_i$. In (ii), we further investigate different shapes, colors, and sizes of visual prompts to draw on top of the image. We will show the effects of different design choices in the experimental section.

**Stage 5: Verification Execution.** After obtaining the region descriptions, the next stage is to verify whether they contain the original entities to assess if any hallucinations exist. Specifically, for each entity $e_i$, given the region descriptions $\mathcal{A}_i = \{a_{i,j}\}, j = 1, ..., L$, where $L$ is the number of sampled answers, we prompt the LLM $L$ times to check whether each $a_{i,j}$ contains $e_i$ and respond with "Yes" or "No". We then map the "Yes-No" response into a binary verification score $v_{i,j}$, where "Yes" is equal to 1 and "No" is equal to 0. The final averaged verification score $V$ for entity $e_i$ is formulated as follows:

$$V\left(e_i\right) = \frac{1}{L} \sum_{j=1}^{L} v_{i,j}. \tag{1}$$

A lower $V(e_i)$ indicates that the entity is likely to be hallucinated. Thus, $V(e_i)$ serves as an indicator for hallucination detection. We determine whether the entity is hallucinated by setting a threshold

Table 1: **Results on POPE.** We report both accuracy and F1 score. Results for LogicCheckGPT are reproduced by us using the official code. Numbers for other methods are taken from Wu et al. (2024). The best results are in **bold**.

| Model | Method | Adversarial | | Popular | | Random | | Average | |
|---|---|---|---|---|---|---|---|---|---|
| | | Acc | F1 | Acc | F1 | Acc | F1 | Acc | F1 |
| mPLUG-Owl | Vanilla | 50.67 | 66.81 | 51.67 | 67.26 | 55.33 | 68.98 | 52.56 | 67.68 |
| | LRV-Instruction | 59.67 | 69.21 | 68.33 | 74.11 | 74.33 | 77.94 | 67.44 | 73.75 |
| | SelfCheck | 66.67 | 74.09 | 72.00 | 77.29 | 70.66 | 75.82 | 69.78 | 75.73 |
| | LURE | 76.33 | 76.72 | 79.67 | 78.75 | 81.33 | 80.95 | 79.11 | 78.81 |
| | LogicCheckGPT | 81.00 | 81.19 | 84.00 | 83.89 | 90.00 | 89.44 | 85.00 | 84.84 |
| | CoReVe (ours) | **83.67** | **83.93** | **85.67** | **85.71** | **91.67** | **91.35** | **87.00** | **87.00** |
| MiniGPT-4 | Vanilla | 72.67 | 75.88 | 78.33 | 79.87 | 84.33 | 84.59 | 78.44 | 80.11 |
| | LRV-Instruction | 74.00 | 71.11 | 80.33 | 78.70 | 81.67 | 80.97 | 78.67 | 76.93 |
| | SelfCheck | 73.00 | 72.72 | 76.67 | 75.86 | 76.00 | 73.53 | 75.22 | 74.04 |
| | LURE | 77.67 | 79.14 | 80.67 | 80.67 | 83.67 | 84.14 | 80.67 | 81.29 |
| | LogicCheckGPT | 80.67 | 78.99 | 82.67 | 80.30 | 84.67 | 82.71 | 82.67 | 80.67 |
| | CoReVe (ours) | **83.33** | **82.64** | **88.00** | **86.76** | **86.33** | **84.98** | **85.89** | **84.79** |
| LLaVA-1.5 | Vanilla | 83.33 | 84.84 | 84.67 | 85.89 | 93.00 | 93.02 | 87.00 | 87.92 |
| | SelfCheck | 88.67 | 88.27 | 88.67 | 88.59 | 90.33 | 89.53 | 89.22 | 88.80 |
| | LURE | 85.33 | 86.25 | 87.00 | 87.05 | 89.67 | 89.70 | 87.33 | 87.67 |
| | LogicCheckGPT | **89.33** | **89.40** | 91.00 | 90.79 | 92.67 | 92.25 | 91.00 | 90.81 |
| | CoReVe (ours) | 88.67 | 88.36 | **92.00** | **91.72** | **94.00** | **93.67** | **91.56** | **91.25** |
| Qwen2.5-VL | Vanilla | 85.67 | 83.65 | 86.33 | 84.29 | 86.67 | 84.62 | 86.22 | 84.19 |
| | LogicCheckGPT | 87.00 | 85.50 | 86.67 | 84.96 | 87.33 | 85.71 | 87.00 | 85.39 |
| | CoReVe (ours) | **87.33** | **86.13** | **87.67** | **86.35** | **89.00** | **88.00** | **88.00** | **86.83** |

$\tau \in [0, 1]$, where $V(e_i) < \tau$ means that entity $e_i$ is hallucinated. The prompt we use in this stage is detailed in Appendix C.2.

**Stage 6: Final Response Generation.** After identifying the hallucinated entities, the final stage is to generate the improved response that takes verification into account. Specifically, we prompt the LLM to revise the initial response based on the verification results in Stage 5. The detailed prompt is listed in Appendix C.3.

## 4 EXPERIMENTS

### 4.1 EXPERIMENTAL SETUP

**Baselines.** We build our CoReVe upon several popular open-source LVLMs, including mPLUG-Owl (mplug-owl-llama-7b) (Ye et al., 2023), LLaVA (llava-v1.5-7b) (Liu et al., 2023), MiniGPT-4 (vicuna-13b) (Zhu et al., 2023), and Qwen2.5-VL (qwen2.5-vl-7b-instruct) (Bai et al., 2025). We refer to the base LVLMs as vanilla. In addition, we also compare CoReVe with other advanced hallucination mitigation methods, including LRV-Instruction (Liu et al., 2024a), SelfCheckGPT (Manakul et al., 2023), LURE (Zhou et al., 2024b), and LogicCheckGPT (Wu et al., 2024). Due to limited space, we provide more implementation details in Appendix A.

**Benchmarks.** We evaluate our CoReVe on several widely used hallucination benchmarks, including POPE (Li et al., 2023b), MME (Fu et al., 2023), CHAIR (Rohrbach et al., 2018), and GPT-4o assisted evaluation. Specifically, POPE and MME are VQA-based benchmarks with binary "Yes-or-No" questions. CHAIR is a captioning-based benchmark, and GPT-4o assisted evaluation is an open-ended image description benchmark. More benchmark details are provided in Appendix B.

### 4.2 MAIN RESULTS

**Results on POPE.** Table 1 presents our results on the POPE benchmark. CoReVe consistently outperforms each LVLM by significant margins regardless of settings. For example, mPLUG-Owl only

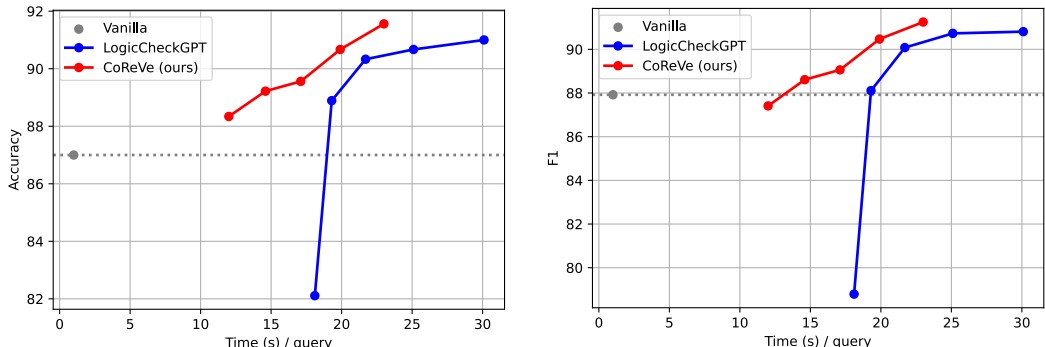

Figure 3: **Trade-off between performance and test-time compute**. The vanilla model is LLaVA-v1.5-7B. We report both accuracy and F1 score averaged across the three splits of POPE. The computational cost is represented as the time (seconds) per query, measured on a single 48GB NVIDIA L40S GPU. CoReVe is more efficient than LogicCheckGPT, always achieving the same performance with less computational cost.

Table 2: **Results on MME existence subset.** We report both accuracy and accuracy+. Results for LogicCheckGPT are reproduced by us using the official code. Numbers for other methods are taken from Wu et al. (2024). The best results are in **bold**.

| Model | Method | Acc | Acc+ |
|---|---|---|---|
| mPLUG-Owl | Vanilla | 65.00 | 35.00 |
| | LRV-Instruction | 83.33 | 66.67 |
| | SelfCheck | 85.00 | 73.33 |
| | LURE | 80.00 | 60.00 |
| | LogicCheckGPT | 95.00 | 90.00 |
| | CoReVe (ours) | **96.67** | **93.33** |
| MiniGPT-4 | Vanilla | 78.33 | 56.67 |
| | LRV-Instruction | 83.33 | 66.67 |
| | SelfCheck | 80.00 | 60.00 |
| | LURE | 85.00 | 70.00 |
| | LogicCheckGPT | 86.67 | 73.33 |
| | CoReVe (ours) | **88.33** | **76.67** |
| LLaVA-1.5 | Vanilla | 96.67 | 93.33 |
| | SelfCheck | 96.67 | 93.33 |
| | LURE | 93.33 | 86.67 |
| | LogicCheckGPT | 96.67 | 93.33 |
| | CoReVe (ours) | **98.33** | **96.67** |
| Qwen2.5-VL | Vanilla | 95.00 | 90.00 |
| | LogicCheckGPT | 95.00 | 90.00 |
| | CoReVe (ours) | **98.33** | **96.67** |

Table 3: **Results on CHAIR.** We report $CHAIR_S$, $CHAIR_I$, and F1 score. Results for LogicCheckGPT are reproduced by us. The best results are in **bold**.

| Model | Method | $CHAIR_S \downarrow$ | $CHAIR_I \downarrow$ | F1 $\uparrow$ |
|---|---|---|---|---|
| mPLUG-Owl | Vanilla | 88.0 | 32.1 | 60.1 |
| | LogicCheckGPT | 70.0 | 25.9 | 61.4 |
| | CoReVe (ours) | **64.0** | **21.7** | **64.3** |
| MiniGPT-4 | Vanilla | 46.0 | 13.0 | 66.0 |
| | LogicCheckGPT | 44.0 | 12.9 | 66.4 |
| | CoReVe (ours) | **34.0** | **9.2** | **67.1** |
| LLaVA-1.5 | Vanilla | 52.0 | 16.2 | 73.2 |
| | LogicCheckGPT | 38.0 | 11.5 | 74.8 |
| | CoReVe (ours) | **30.0** | **8.7** | **77.5** |
| Qwen2.5-VL | Vanilla | 42.0 | 8.3 | 73.0 |
| | LogicCheckGPT | 40.0 | 8.2 | 73.6 |
| | CoReVe (ours) | **30.0** | **6.9** | **74.4** |

Table 4: **Results on GPT-4o assisted evaluation.** We report both accuracy and relevancy scores. The best results are in **bold**.

| Model | Method | Acc | Rel |
|---|---|---|---|
| mPLUG-Owl | Vanilla | 4.78 | 7.94 |
| | CoReVe (ours) | **6.91** | **8.41** |
| MiniGPT-4 | Vanilla | 5.69 | 8.62 |
| | CoReVe (ours) | **7.24** | **8.90** |
| LLaVA-1.5 | Vanilla | 6.48 | 8.91 |
| | CoReVe (ours) | **7.48** | **9.03** |
| Qwen2.5-VL | Vanilla | 8.35 | 9.65 |
| | CoReVe (ours) | **8.76** | **9.72** |

achieves an average of 52.56% accuracy and 67.68% F1 score across the three splits. By incorporating CoReVe, mPLUG-Owl can achieve +34.44% accuracy and +19.32% F1 score improvements. For MiniGPT-4, CoReVe yields +7.45% accuracy and +4.68% F1 score gains. When building upon a stronger LLaVA-1.5 (Qwen2.5-VL), we still observe +4.56% (+1.78%) and +3.33% (+2.64%) improvements on accuracy and F1 score, respectively. This demonstrates the effectiveness and the generality of our method.

To further demonstrate the superiority of CoReVe, we analyze the trade-off between performance and test-time compute. As shown in Figure 3, compared with the prior state of the art, *i.e.*, Logic-CheckGPT, our CoReVe can always achieve the same performance with less computational cost. It

Table 5: **Ablations for CoReVe on POPE.** The baseline LVLM is LLaVA-v1.5-7B. We report both accuracy and F1 score averaged across the three splits. Unless otherwise specified, the default settings are: (a) the number of sampled answers is 7, (b) the hallucination threshold is 0.1, (c) the image prompt is the image overlaid with the bounding box, (d) the shape of the bounding box is rectangle, (e) the color of the bounding box is red, and (f) the size of the bounding box is 1 pixel. The default entry is marked in  gray .

(a) **Number of answers.** A moderate increase in the number of sampled answers performs better.

| $L$ | Acc | F1 |
|---|---|---|
| - | 87.00 | 87.92 |
| 3 | 88.34 | 87.41 |
| 5 | 89.56 | 89.06 |
| 7 | **91.56** | **91.25** |
| 9 | 90.67 | 90.49 |

(b) **Hallucination threshold.** A lower threshold works the best.

| $\tau$ | Acc | F1 |
|---|---|---|
| - | 87.00 | 87.92 |
| 0.1 | **91.56** | **91.25** |
| 0.2 | 88.78 | 87.79 |
| 0.3 | 87.00 | 85.51 |
| 0.4 | 87.67 | 86.44 |

(c) **Image prompt.** Drawing the bounding box on the image leads to more gains.

| Type | Acc | F1 |
|---|---|---|
| - | 87.00 | 87.92 |
| original | 89.67 | 89.36 |
| overlaid | **91.56** | **91.25** |
| cropped | 83.89 | 81.53 |

(d) **Bounding box shape.** A rectangular shape is more effective.

| Shape | Acc | F1 |
|---|---|---|
| - | 87.00 | 87.92 |
| rectangle | **91.56** | **91.25** |
| incircle | 90.11 | 89.87 |
| circumcircle | 89.56 | 89.39 |

(e) **Bounding box color.** A red color yields better performance.

| Color | Acc | F1 |
|---|---|---|
| - | 87.00 | 87.92 |
| red | **91.56** | **91.25** |
| green | 90.22 | 89.89 |
| blue | 90.11 | 89.68 |
| white | 90.67 | 90.40 |

(f) **Bounding box size.** Using a 1-pixel size is enough.

| Size | Acc | F1 |
|---|---|---|
| - | 87.00 | 87.92 |
| 1 | **91.56** | **91.25** |
| 2 | 90.00 | 89.62 |
| 3 | 89.55 | 89.15 |
| 4 | 89.34 | 88.90 |

is worth noting that it takes at least 19 seconds per query for LogicCheckGPT to start to surpass the performance of the vanilla model, whereas ours only takes around 13 seconds. This demonstrates the efficiency of our method.

**Results on MME Existence Subset.** We also evaluate our method on the MME existence subset, focusing on the object existence hallucination. The results are shown in Table 2. Again, our CoReVe consistently outperforms each LVLM by clear margins. Specificaly, we achieve +31.67% accuracy, +58.33% accuracy+ gains for mPLUG-Owl, +10.00% accuracy, +20.00% accuracy+ gains for MiniGPT-4, +1.66% accuracy, +3.34% accuracy+ gains for LLaVA-1.5, and +3.33% accuracy, +6.67% accuracy+ gains for Qwen2.5-VL, respectively. Note that LLaVA-1.5 and Qwen2.5-VL already achieve very strong performances under this benchmark. Nevertheless, our CoReVe can still improve upon them.

**Results on CHAIR.** Apart from binary "Yes-or-No" questions in POPE and MME, we further evaluate our method on CHAIR, which is a more challenging hallucination benchmark as it involves multiple objects in the captions. As shown in Table 3, our CoReVe significantly reduces the CHAIR hallucination scores for all LVLMs, while also improving their F1 scores. In addition, our CoReVe also surpasses LogicCheckGPT, setting a new state of the art in this more challenging benchmark.

**Results on GPT-4o Assisted Evaluation.** Going beyond CHAIR, we employ GPT-4o for more comprehensive open-ended evaluation. The results are shown in Table 4. Our CoReVe provides more accurate responses on all LVLMs while also enhancing the relevancy of the generated descriptions. Given that the visual understanding and language logic capabilities of GPT-4o have already reached a human level, it can more comprehensively assess the benefits brought by our method.

## 4.3 ABLATION STUDY

We ablate our CoReVe using LLaVA-v1.5-7B as the default LVLM on the POPE benchmark. Several intriguing properties are observed.

**Sampled Answer Numbers.** We first study the effect of the number of sampled answers $L$ per region in the region description stage. As shown in Table 5a, sampling three answers per region has

Table 6: **Effect of using the LVLM alone.** The vanilla LVLM is LLaVA-v1.5-7B. We report both accuracy and F1 score averaged across the three splits of POPE. CoReVe can handle all stages using the LVLM alone.

| Method | Acc | F1 |
|---|---|---|
| Vanilla | 87.00 | 87.92 |
| LogicCheckGPT (w/ GPT) | 91.00 | 90.81 |
| CoReVe (w/ GPT) | **91.56** | **91.25** |
| LogicCheckGPT (w/o GPT) | 83.33 | 81.01 |
| CoReVe (w/o GPT) | **89.89** | **89.11** |

Table 7: **Effect of using ground truth bounding boxes.** The vanilla LVLM is LLaVA-v1.5-7B. We report $CHAIR_S$, $CHAIR_I$, and F1 score on CHAIR. More accurate bounding boxes are beneficial for truly existent objects but not applicable to hallucinated (*i.e.*, nonexistent) ones.

| Method | $CHAIR_S \downarrow$ | $CHAIR_I \downarrow$ | F1 $\uparrow$ |
|---|---|---|---|
| Vanilla | 52.0 | 16.2 | 73.2 |
| CoReVe (default) | 30.0 | 8.7 | **77.5** |
| CoReVe (w/ GT) | **24.0** | **7.0** | 76.7 |

already yielded +1.34% accuracy gains. Increasing the number of sampled answers further improves the performance. It makes sense as generating more answers tends to increase the diversity of region descriptions, which is helpful for the subsequent verification stage. The best performance is achieved by setting $L = 7$. The performance tends to saturate when $L$ is increased further.

**Hallucination Threshold.** We then study the effect of the hallucination threshold $\tau$ in the verification execution stage. The results are shown in Table 5b. The model already reaches its performance peak by setting $\tau = 0.1$. We observe a noticeable performance drop when continuing to increase the threshold. A higher threshold tends to misclassify a large number of existent objects as nonexistent, thus causing a drastic increase in false negatives.

**Image Prompt Type.** Table 5c studies the types of image prompts to use in the region description stage. The results demonstrate that using the image content overlaid with the generated bounding box leads to better performance than using the original image. Apart from the text prompt, using such a visual prompt better instructs the model to focus on the region of interest when producing answers. Another naïve way to utilize the local information is to crop the image with the generated bounding box coordinates. However, we observe that simply cropping the image significantly degrades the performance due to the loss of the image context.

**Bounding Box Shape.** Given the bounding box coordinates, how to draw the shape on top of the image is worth exploring. We consider three possible shapes with the provided coordinates: rectangle, incircle, and circumcircle. As shown in Table 5d, all shapes can significantly surpass the baseline, where a standard rectangular shape is the most effective choice.

**Bounding Box Color.** We compare different bounding box colors to draw in Table 5e. All examined colors outperform the baseline by a clear margin, demonstrating the robustness of our method. A red color stands out and yields the best performance. This phenomenon could be related to the distribution of the datasets used to train the model.

**Bounding Box Size.** Table 5e ablates the effect of the bounding box size to draw on the image. The results indicate that using a 1-pixel size is enough to achieve the best performance. A larger size tends to decrease the performance as it may potentially obscure the content presented in the image.

## 4.4 FURTHER ANALYSIS

**Effect of Using the LVLM Alone.** In our default setting, we adopt an LLM (*e.g.*, GPT-3.5) to facilitate the stages that only involve processing the generated text, *i.e.*, entity extraction, verification execution, and final response generation. This raises the question of whether the LVLM itself can perform all the stages, given that an expert model may not always be available in practice. We verify this using LLaVA-v1.5-7B. The results on POPE are shown in Table 6. Using the model alone can also surpass the vanilla model by a clear margin, improving the accuracy by 2.89%, indicating that GPT is not indispensable in CoReVe. In contrast, without the help of an LLM, the performance of LogicCheckGPT degrades significantly, even underperforming the vanilla model. We attribute this to LogicCheckGPT's heavy reliance on GPT to construct the logical closed loop (*e.g.*, attribute-to-object inquiring). This further demonstrates the promise of our method in reducing hallucinations without relying on any other models. We also investigate the failure cases of using the model itself and find that most errors arise from entity extraction and consistency checking.

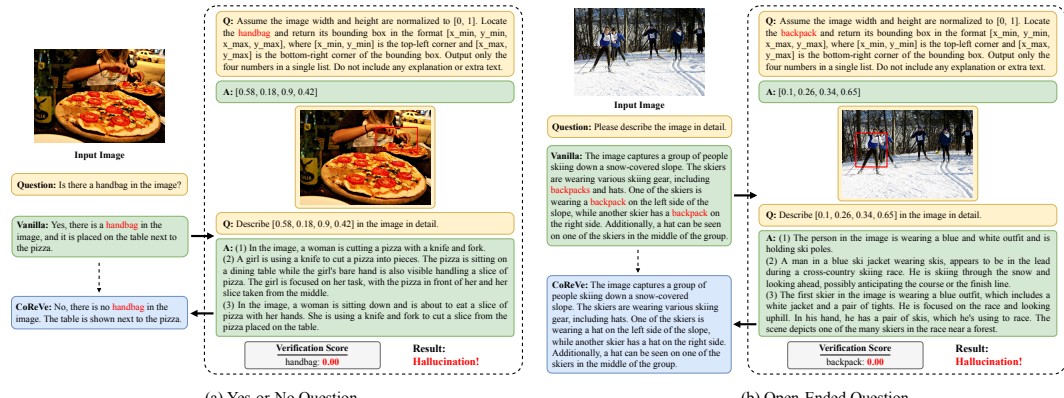

Figure 4: **Example results of CoReVe for LLaVA-1.5.** The hallucinated objects are highlighted in red. We sample three region descriptions per examinee to produce a set of diverse answers for verification. More examples are provided in Appendix D.

These errors primarily originate from inherent limitations within the model and prompt design. We expect that a better design of prompts may further unlock the potential of using the model itself for hallucination mitigation. We leave it for future work.

**Effect of Using Ground Truth Bounding Boxes.** Our CoReVe relies on the LVLM itself to generate bounding box coordinates. It is interesting to ask: what if such bounding boxes for objects are already available and perfect? We investigate this on the CHAIR benchmark, considering it is more open-ended and contains multiple objects in the images and captions. Specifically, we use ground truth annotations from the COCO 2014 validation set (Lin et al., 2014). For each query, after extracting candidate objects from the initial response, we replace the bounding boxes generated by the LVLM with ground truth ones for those truly existent objects in the image, while keeping all other procedures unchanged. The results are shown in Table 7. Using ground truth bounding boxes further reduces CHAIR hallucination scores, with a slight decrease in the F1 score. This indicates that more accurate localization benefits truly existent objects. However, ground truth bounding boxes are, of course, not available for nonexistent objects. Thus, we still have to rely on pseudo bounding boxes from the LVLM for verification. We expect that the performance of CoReVe can be further improved when LVLMs get better at localizing objects.

**Qualitative Examples.** We present two representative examples covering two types of questions: (a) yes-or-no, and (b) open-ended, using LLaVA-v1.5-7B as an exemplar LVLM. As shown in Figure 4, CoReVe successfully detects the hallucinated objects "handbag" and "backpack"—both assigned a verification score of 0.00—and corrects the initial responses accordingly. By applying chain-of-region verification, we make the hallucination mitigation process more interpretable—the self-generated bounding boxes help explain how hallucinations occur. For instance, when asked to localize "handbag" and "backpack", the LVLM instead outputs regions corresponding to "hand" and "skier", respectively, which commonly co-occur in similar visual contexts. When prompted with the image containing the bounding boxes, the LVLM tends to refocus on the regions of interest, where "handbag" and "backpack" no longer appear. Such a process cannot be simply achieved by external object detection models or manual annotations—we cannot provide bounding boxes for the hallucinated objects as they do not exist. This further demonstrates the superiority of our method.

## 5 CONCLUSION

In this work, we have introduced CoReVe, a visual chain-of-verification method that triggers region-level processing from LVLMs themselves to mitigate their own hallucinations. CoReVe mimics how humans comprehend intricate visual information by delving into details in specific image regions. Our method can be seamlessly applied to various LVLMs without retraining or relying on external detection models. Extensive experiments on POPE, MME, CHAIR, and GPT-4o assisted evaluation demonstrate the superiority of our method in both performance and efficiency. We hope our explorations can pave the way for more efficient, reliable, and interpretable hallucination mitigation.

ETHICS STATEMENT

This work addresses the critical object hallucination issue to enhance the reliability and trustworthiness of LVLMs. The proposed method reduces the risks of misinformation and biased content, which advances the responsible development of artificial intelligence (AI) systems and promotes greater public trust in AI technologies.

REPRODUCIBILITY STATEMENT

We provide detailed hyperparameter specifications for our experiments in the main text (Section 4) and the supplementary material (Appendices A–C) to ensure reproducibility. We will release the code to the research community for reproducible research.

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

## A    MORE IMPLEMENTATION DETAILS

We conduct our experiments using an NVIDIA L40S GPU (48GB) and an AMD EPYC 7702 CPU. All implementations are based on the PyTorch framework (Paszke et al., 2019), incorporating components from the HuggingFace Transformers library (Wolf et al., 2019). Unless otherwise specified, we maintain the default hyperparameter settings for all LVLMs. Specifically, we set the number of sampled answers $L = 7$ for region description, and the hallucination threshold $\tau = 0.1$ for verification execution. We adopt the image overlaid with the bounding box as the visual prompt, where the shape of the bounding box is a rectangle, the color of the bounding box is red, and the size of the bounding box is 1 pixel. In addition, we employ GPT-3.5 Turbo (gpt-3.5-turbo-0125) as the default LLM to help process the generated text for entity extraction, verification execution, and final response generation.

Note that we employ GPT-3.5 only for a fair comparison with the prior state of the art, *i.e.*, LogicCheckGPT (Wu et al., 2024). We have shown in the main text that our CoReVe can still achieve promising results without relying on any external models, which cannot be achieved by previous methods. It is also worth mentioning that using the same hyperparameters across different LVLMs can be suboptimal. Nevertheless, CoReVe outperforms each baseline regardless of models, tasks, and settings.

## B    MORE BENCHMARK DETAILS

### B.1    POPE

POPE (Li et al., 2023b) is a hallucination evaluation benchmark designed in the VQA paradigm. Specifically, it evaluates hallucinations by querying LVLMs with the questions in the form of "`Is there a/an <object> in the image?`", where `<object>` is selected from three different types of splits: random, popular, and adversarial. For the "random" split, objects are taken randomly from the entire dataset. For the "popular" split, objects are chosen from the most frequent object list. For the "adversarial" split, objects that are highly related to the image objects are selected. We conduct our evaluation on the COCO 2014 validation set (Lin et al., 2014). Following Wu et al. (2024), for each split, we sample 50 images and 6 questions for each image, resulting in a total of 300 questions that are evenly distributed between positive and negative samples (50%-50%). We evaluate the final performance using both accuracy and F1 score.

### B.2    MME

MME (Fu et al., 2023) is a VQA-based benchmark for evaluating the perceptual and cognitive capabilities of LVLMs across a wide range of subtasks. Following Wu et al. (2024), we use the existence subset for object-level hallucination evaluation. Specifically, similar to POPE, MME consists of two binary "Yes-or-No" questions for each image in the subset. We adopt accuracy and accuracy+ as the evaluation metrics. The former is calculated based on each question, while the latter is calculated based on each image, requiring both questions to be answered correctly.

### B.3    CHAIR

Apart from binary "Yes-or-No" questions, we adopt CHAIR (Rohrbach et al., 2018) to evaluate hallucinations in the image captioning task. Specifically, CHAIR operates by computing the proportion of all objects mentioned in the caption that do not exist in the ground-truth annotations. CHAIR comprises two main metrics, including $\text{CHAIR}_\text{I}$ and $\text{CHAIR}_\text{S}$ that assess instance-level and sentence-level hallucinations, respectively. They are formulated as follows:

$$\text{CHAIR}_\text{I} = \frac{|\{\text{hallucinated objects}\}|}{|\{\text{all mentioned objects}\}|},$$
$$\text{CHAIR}_\text{S} = \frac{|\{\text{captions w/ hallucinated objects}\}|}{|\{\text{all captions}\}|}, \tag{2}$$

where lower values indicate fewer hallucinations. Besides, we also consider the F1 score to assess the richness and the accuracy of the generated captions.

Table 8: **Prompt template for entity extraction.** {In-context examples} are in-context examples for better instruction. {Input sentence} is the output from the initial response generation stage.

---

**System prompt**
You are a language assistant that helps to extract information from given sentences.

---

**Prompt**
You are given a sentence, extract the entities within the sentence for me.

[Task]
Your task is to extract the common objects and summarize them as general categories without repetition, merging essentially similar objects. Avoid extracting abstract or non-specific entities. Extract entity in the singular form. Output all the extracted types of items in one line and separate each object type with a period. If there is nothing to output, then output a single "None". DO NOT RESPOND WITH ANYTHING ELSE.

Here are examples:
{In-context examples}

Now complete the following:

[Sentence]
{Input sentence}

[Response]

---

Table 9: **Prompt template for verification execution.** {Input statement} is the output from the region description stage. {object} is the entity to be verified based on the statement.

---

**System prompt**
You are a language assistant that helps to answer the question according to instructions.

---

**Prompt**
You are given a statement and a question.

[Task]
Your task is to answer the question based on the statement. The statement is about some objects. The question is to ask whether some specific object exists.
1. Your response should be limited to one of the following two choices: "Yes"/"No".
2. Note that instances of a certain category can also belong to its super-categories. For example, a baseball is a subclass of the sports ball.
3. Note that the table is equivalent to the dining table here.
4. DO NOT RESPOND WITH ANYTHING ELSE.

[Response Format]
Yes/No

Now complete the following:

[Statement]
{Input statement}

[Question]
Is there a {object} in the statement?

[Response]

---

## B.4 GPT-4O ASSISTED EVALUATION

To further evaluate the effectiveness of our method in image description tasks, we go beyond CHAIR metrics and adopt GPT-4o for open-ended evaluation. Specifically, following the common proto-

Table 10: **Prompt template for final response generation.** {In-context examples} are in-context examples for better instruction. {Input query} is the question asked by the user. {Input passage} is the initial response to be corrected. {Input information} is the supplementary information from the verification execution stage.

---

**System prompt**
You are a language assistant that helps to refine a passage according to instructions.

---

**Prompt**
You are given a query, a passage and supplementary information.

[Task]
You are required to correct and output the refined passage in a fluent and natural style, following these rules:
1. Correct the sentences in the passage if they are inconsistent with the supplementary information. Remove the objects that are confirmed to not exist in the supplementary information.
2. Do not modify correct sentences and introduce additional information.
3. When giving refined passage, also pay attention to the given query. The refined passage should be a reasonable answer to the query.
4. Note the dining table is equivalent to the table.
Output only the corrected passage, without introducing extra contents.

Here are examples:
{In-context examples}

Now complete the following:

[Query]
{Input query}

[Passage]
{Input passage}

[Supplementary Information]
{Input information}

[Response]

---

col (Liu et al., 2024a; Yin et al., 2024; Wu et al., 2024), we sample 500 images from the COCO 2014 validation set and ask the model to generate detailed descriptions. Afterward, we prompt GPT-4o to score the original response and our response based on the instruction and the image. GPT-4o evaluation takes into account two dimensions: Accuracy and Relevancy. More detailed prompt construction is provided in Section C.4.

## C   PROMPT TEMPLATES

In this section, we provide the detailed prompt templates used in the following stages: entity extraction, verification execution, and final response generation. In addition, we provide the detailed prompt template used for GPT-4o assisted evaluation.

### C.1   ENTITY EXTRACTION

The prompt template for entity extraction is detailed in Table 8.

### C.2   VERIFICATION EXECUTION

The prompt template for verification execution is detailed in Table 9.

Table 11: **Prompt template for GPT-4o assisted evaluation.** {Response of Assistant 1} and {Response of Assistant 2} are the initial response and the final response, respectively.

---

**System prompt**
You are required to score the performance of two AI assistants in describing a given image.

**Prompt**
You should pay extra attention to the hallucination, which refers to the part of descriptions that are inconsistent with the image content, such as claiming the existence of something not present in the image or describing incorrectly in terms of the counts, positions, or colors of objects in the image. Please rate the responses of the assistants on a scale of 1 to 10, where a higher score indicates better performance, according to the following criteria:
1: Accuracy: whether the response is accurate with respect to the image content. Responses with fewer hallucinations should be given higher scores.
2: Relevancy: whether the response directly follows the instruction.
Please output the scores for each criterion, containing only two values indicating the scores for Assistant 1 and 2, respectively. The two scores are separated by a space. Following the scores, please provide an explanation of your evaluation, avoiding any potential bias and ensuring that the order in which the responses were presented does not affect your judgment.

[Assistant 1]
{Response of Assistant 1}
[End of Assistant 1]

[Assistant 2]
{Response of Assistant 2}
[End of Assistant 2]

Output format:

Accuracy: <Scores of the two answers>
Reason:

Relevancy: <Scores of the two answers>
Reason:

---

### C.3 FINAL RESPONSE GENERATION

The prompt template for final response generation is detailed in Table 10.

### C.4 GPT-4O ASSISTED EVALUATION

The prompt template for GPT-4o assisted evaluation is detailed in Table 11.

## D MORE QUALITATIVE RESULTS

In this section, we present more qualitative results, including two types of questions: (i) yes-or-no questions, and (ii) open-ended questions. We use LLaVA-v1.5-7B as an exemplar LVLM for visualization.

### D.1 YES-OR-NO QUESTIONS

Figure 5 provides more examples with two yes-or-no questions: "*Is there a handbag in the image?*" and "*Is there a truck in the image?*". Our CoReVe assigns a verification score of 0.67 to the existent object 'handbag' and 0.00 to the hallucinated object "truck", respectively. Since the score for "truck" is lower than our pre-defined threshold of 0.1, CoReVe effectively identifies the "truck" as the hallucinated object and corrects the initial response accordingly.

For existent objects, the bounding box coordinates generated from the LVLM provide a rough location for them, which better instructs the LVLM to focus on specific image regions before gen-

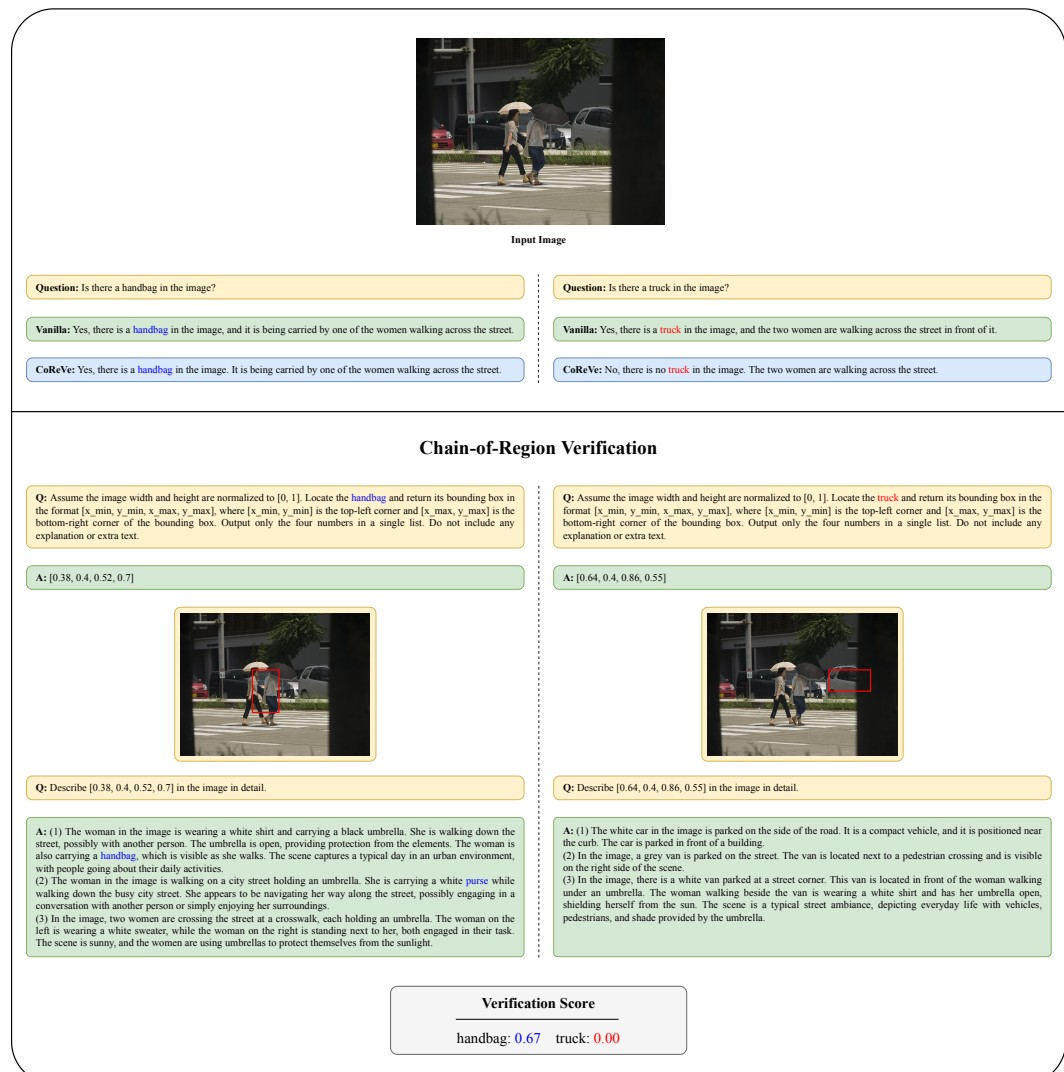

Figure 5: **Example results of CoReVe for yes-or-no questions.** The vanilla LVLM is LLaVA-v1.5-7B. The existent objects are highlighted in blue, while the hallucinated objects are highlighted in red. We sample three region descriptions per examinee to produce a set of diverse answers for verification.

erating the final answer. For nonexistent objects, the generated bounding box coordinates are still meaningful—they explain how the LVLM hallucinates to some extent. For example, when asked to provide the bounding box coordinates for the "truck", the LVLM actually outputs the location of a car in the background. This indicates that the LVLM mistakes the "car" for a "truck". After being prompted with the image containing the bounding box, the LVLM tends to refocus on the region of interest, where the "truck" no longer appears. This process also makes the response from the LVLM more interpretable, as it provides cues about how the model derives its final answer.

## D.2 OPEN-ENDED QUESTIONS

Figure 6 provides more examples with an open-ended question: "*Please describe the image in detail.*" As the length of the generated response increases, the LVLM tends to hallucinate more nonexistent objects, causing a more severe hallucination issue. Nevertheless, our CoReVe successfully detects the hallucinated objects "chair", "potted plants", "clock", and "vase", all of which have a verification score of 0.00.

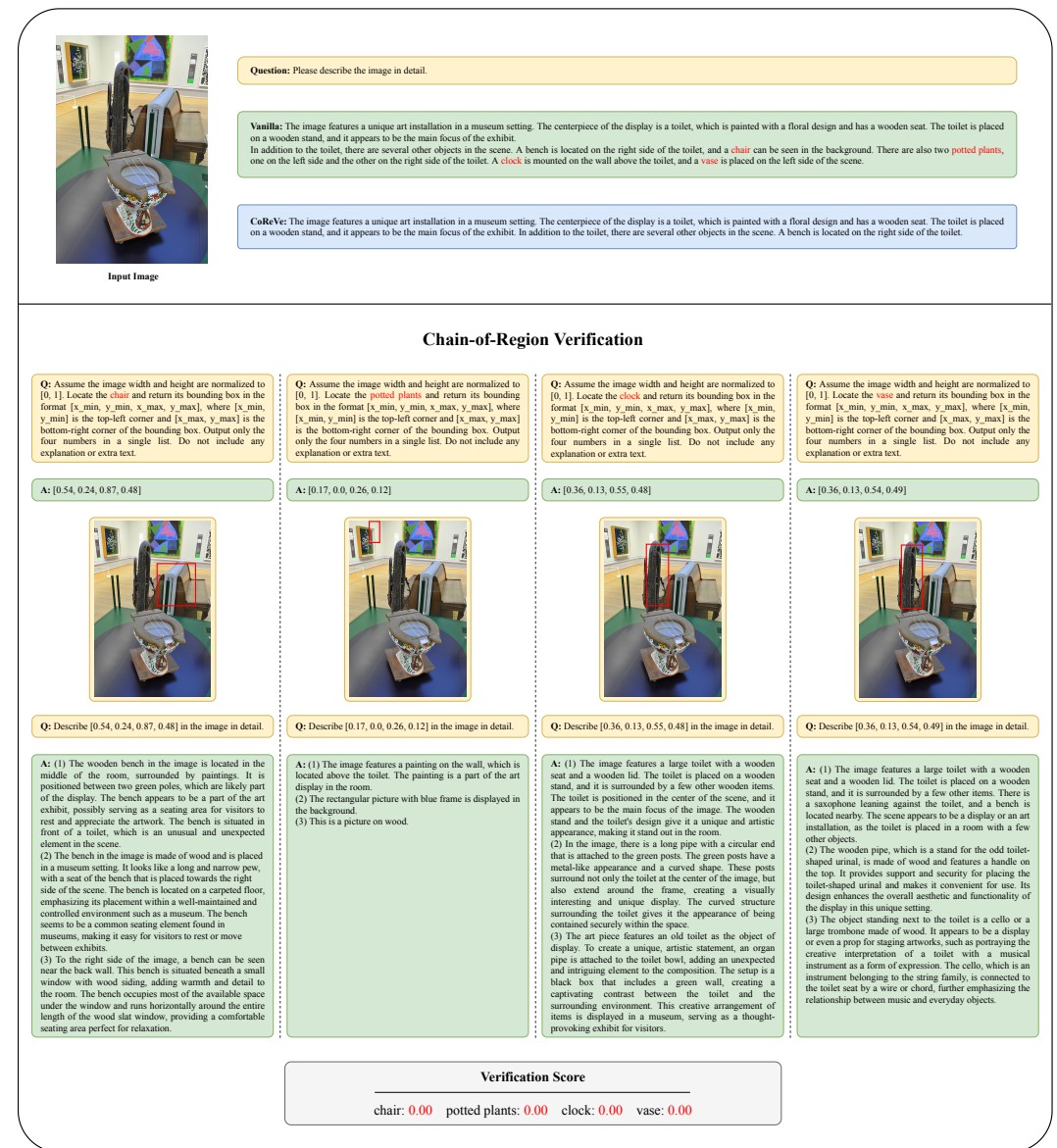

Figure 6: **Example results of CoReVe for open-ended questions.** The vanilla LVLM is LLaVA-v1.5-7B. The hallucinated objects are highlighted in red. We sample three region descriptions per examinee to produce a set of diverse answers for verification.

Similar to the cases of yes-or-no questions, such a chain-of-region verification further enhances the interpretability of the hallucination mitigation process. More specifically, with the aid of the generated bounding boxes and region descriptions, one can easily understand that the LVLM misidentifies the bench as a "chair", the painting as "potted plants", the musical-instrument-shaped pipe as a "clock" or a "vase", respectively. Based on this information, the LVLM can finally correct its initially hallucinated response.

