# OpenReview forum: "CoReVe: Mitigating Object Hallucinations in Large Vision-Language Models via Chain-of-Region Verification"
_ICLR.cc/2026/Conference — ICLR 2026 Conference Withdrawn Submission_

### Official Review · Reviewer_Kvq1 · 2025-10-17

**Soundness:** 2
**Presentation:** 3
**Contribution:** 2
**Rating:** 4
**Confidence:** 5

**Summary:**

The paper proposes CoReVe, a training-free, post-hoc pipeline to reduce object hallucinations in LVLMs. It (1) gets the model’s initial answer, (2) extracts mentioned entities, (3) asks the LVLM to localize each entity with a box, (4) elicits region descriptions for those boxes, (5) verifies mentions via aggregated Yes/No checks, and (6) rewrites the answer to remove hallucinations. On POPE, MME-existence, CHAIR, and a GPT-4o–assisted eval, CoReVe improves accuracy/F1 over vanilla models and LogicCheckGPT with reasonable compute. Strengths are plug-and-play usability and interpretability via explicit regions; open issues include reliance on self-generated boxes, limited coverage beyond object existence (attributes/relations), and the need for stronger robustness/statistical validation.

**Strengths:**

Training-free and model-agnostic: CoReVe plugs into diverse LVLMs without finetuning or external detectors, yielding immediate, practical gains.

Region-level interpretability: the chain-of-region process (self-localization → local descriptions → verification) makes corrections transparent and auditable at the object/region level.

Solid empirical evidence: consistent improvements over vanilla and prior post-hoc baselines across POPE, MME-existence, CHAIR, and GPT-4o–assisted evaluation, with favorable accuracy/F1 vs. inference-time trade-offs.

**Weaknesses:**

- The ideas are not novel: the “extract information first, then verify” pipeline has already been proposed by prior work such as Woodpecker: Hallucination Correction for Multimodal Large Language Models, FaithScore: Fine-grained Evaluations of Hallucinations in Large Vision-Language Models, and Unified Hallucination Detection for Multimodal Large Language Models.
- The paper is also missing citations to several multimodal hallucination benchmarks, including HallusionBench: An Advanced Diagnostic Suite for Entangled Language Hallucination and Visual Illusion in Large Vision-Language Models, FaithScore: Fine-grained Evaluations of Hallucinations in Large Vision-Language Models, Holistic Analysis of Hallucination in GPT-4V(ision): Bias and Interference Challenges, FIHA: Autonomous Hallucination Evaluation in Vision-Language Models with Davidson Scene Graphs, and Hallucination-Augmented Contrastive Learning for Multimodal Large Language Models.
- In addition, performance should be compared against training-based methods, such as Aligning Large Multimodal Models with Factually Augmented RLHF, FGAIF: Aligning Large Vision-Language Models with Fine-Grained AI Feedback, and Mitigating Object Hallucination in Large Vision-Language Models with Human-Free Reinforcement Learning. The paper also omits some training-free baselines, for example Woodpecker: Hallucination Correction for Multimodal Large Language Models and A Unified Hallucination Mitigation Framework for Large Vision-Language Models.

**Questions:**

See weakness

---

### Official Review · Reviewer_Jd95 · 2025-10-31

**Soundness:** 2
**Presentation:** 2
**Contribution:** 2
**Rating:** 2
**Confidence:** 5

**Summary:**

The paper addresses the persistent problem of object hallucinations in Large Vision-Language Models (LVLMs)—where models generate descriptions of objects not present in the input image. To mitigate this issue, the authors propose Chain-of-Region Verification (CoReVe), a training-free, post-hoc framework that leverages region-aware visual verification inspired by human visual reasoning. CoReVe operates through a six-step pipeline. By grounding each mentioned entity in specific image regions and verifying its presence before final output, CoReVe effectively reduces hallucinated content.

**Strengths:**

1. The proposed CoReVe method is well-motivated and demonstrates a reasonable approach to mitigating hallucination in vision-language models.
2. The experimental evaluation is comprehensive, covering multiple benchmarks and ablation studies that support the paper’s claims.
3. The manuscript is clearly written and well-structured, making it easy to follow the technical contributions and experimental pipeline.

**Weaknesses:**

1. The paper exhibits similarities—both in phrasing and experimental design—to the prior work LogicCheckGPT (e.g., lines 72–78 and 87–91). The overlap in presentation and methodology detracts from the perceived novelty. The authors are encouraged to rephrase these sections and more clearly differentiate their contributions.
2. Although the method incorporates grounding, the paper does not evaluate the accuracy of the predicted bounding boxes. This is particularly relevant given that many early vision-language models (e.g., mPlug-Owl, MiniGPT-4, LLaVA-1.5) lack explicit grounding capabilities. Moreover, the Image Prompt experiment shows that “overlaid” regions outperform “cropped” ones—could this advantage stem from inaccuracies in grounding? A quantitative analysis of grounding quality would strengthen the argument.
3. In Stage 4 (Region Description), it remains unclear whether the model reliably localizes and describes the specified {coordinate} regions. The authors should provide quantitative metrics (e.g., region-description alignment scores or human evaluation) to validate the fidelity of this critical step.
4. Together, weaknesses 2 and 3 weaken the evidence for CoReVe’s effectiveness, as unverified grounding and region description may compromise its hallucination mitigation claims.

**Questions:**

See the weakness.

---

### Official Review · Reviewer_CK3p · 2025-10-31

**Soundness:** 3
**Presentation:** 3
**Contribution:** 2
**Rating:** 4
**Confidence:** 3

**Summary:**

The paper introduces CoReVe, a training-free method to reduce object hallucinations by eliciting region-level evidence and verifying it. The pipeline starts from the LVLM’s initial answer, extracts entities, elicits bounding boxes for each entity, probes the model to generate region descriptions, runs a consistency check, and rewrites the answer. The design is simple and post-hoc. Results show gains over vanilla LVLMs and recent baselines on POPE, MME, and CHAIR. Ablations probe the design choices behind region prompting. Ablations probe region-prompt design and sampling: overlaying the predicted box on the original image outperforms using the original image or a cropped patch, and sampling a small number of region descriptions per box improves POPE performance. The paper also looks at speed and deployment. On POPE, CoReVe matches or beats LogicCheckGPT in accuracy and F1 while running faster per image on one L40S GPU, which makes the cost vs quality trade-off clear. The LVLM-only mode (no external LLM) stays competitive on POPE, but the best results use the GPT-assisted setup.

**Strengths:**

- CoReVe is a simple chain-of-region verification pipeline that requires no retraining and no external detection models, while explicitly grounding checks in image regions.
- CoReVe works across multiple LVLMs (mPLUG-Owl, MiniGPT-4, LLaVA-1.5, Qwen2.5-VL) with consistent accuracy/F1 gains, indicating good generality.
- Compared to LogicCheckGPT, CoReVe reaches equal or better POPE performance at lower wall-clock .

**Weaknesses:**

- The baseline set omits several training-free, LLM-free decoding controls, which makes the deployment trade-off less clear. It would strengthen the paper to add comparisons with contrastive decoding variants (VCD, ICD, etc). Such additions would give a more comprehensive view of when to prefer this region-verification pipeline over lightweight decoding controls in practical settings.
- The strongest reported results still rely on an external LLM for entity identification and related text-only checks (entity detection and , so the pipeline is not fully self-contained. It would help to report a side-by-side comparison of the LVLM-only variant versus the default configuration. An analysis of failure modes without the external LLM would clarify when the dependency is essential and when the pipeline is truly plug-and-play.
- Verification of inconsistencies in object detection depends on correctly finding entities and boxes, and the cost grows with the number of objects; dense scenes could hurt both accuracy and speed.

**Questions:**

- If the image contains three spoons scattered on a counter but the initial caption in Stage 1 mentions only “a spoon,” how does CoReVe ensure that Stage 3’s coordinate prediction refers to the correct spoon instance? Does the pipeline return one box per entity type or multiple boxes per instance, and how are co-referent mentions disambiguated? More broadly, does the presence of multiple unmentioned instances affect coverage (e.g., omitting additional spoons) or introduce errors in the final rewrite?
- CoReVe processes entities independently with L sampled region descriptions each. Can you report performance and latency as a function of entities-per-image and discuss any mechanisms (deduplication, overlap handling, batched verification) that keep the pipeline efficient and reliable in dense scenes?

---

### Official Review · Reviewer_XLCK · 2025-11-01

**Soundness:** 3
**Presentation:** 3
**Contribution:** 2
**Rating:** 4
**Confidence:** 4

**Summary:**

This paper proposes CoReVe, a training-free, region-aware method to reduce object hallucinations in LVLMs. Specifically, CoReVe enables LVLMs to verify its own predictions by first locating mentioned entities in LVLMs' responses, then describing the regions of the extracted entities and revising its answers based on the verification. Experiments on several benchmarks with multiple LVLMs show that CoReVe consistently lowers hallucination rates in a training-free manner.

**Strengths:**

1. The proposed training-free and plug-and-play method is easy to use and can mitigate object hallucination in LVLMs.
2. The motivation and the logic flow of this paper are clear.

**Weaknesses:**

1. The whole method is based on different prompts (heuristic rules), and the results may be biased since different LVLMs may have different sensitivity to various prompts. It will be better if the authors can have further discussions on the effect of different prompt design.
2. The evaluation benchmark is a little bit outdated, and it will be interesting if the authors can test their method on more hallucination types(i.e., relation hallucination).
3. No significance tests/ statistical analysis are included in the experiments, which are important in evaluating method performance.

**Questions:**

Please see my review above.

---

### Note · Authors · 2025-11-14

I have read and agree with the venue's withdrawal policy on behalf of myself and my co-authors.